# Hepatitis B Virus in West African Children: Systematic Review and Meta-Analysis of HIV and Other Factors Associated with Hepatitis B Infection

**DOI:** 10.3390/ijerph20054142

**Published:** 2023-02-25

**Authors:** Djeneba B. Fofana, Anou M. Somboro, Mamoudou Maiga, Mamadou I. Kampo, Brehima Diakité, Yacouba Cissoko, Sally M. McFall, Claudia A. Hawkins, Almoustapha I. Maiga, Mariam Sylla, Joël Gozlan, Manal H. El-Sayed, Laurence Morand-Joubert, Robert L. Murphy, Mahamadou Diakité, Jane L. Holl

**Affiliations:** 1Faculty of Medicine, University of Sciences, Techniques and Technologies of Bamako (USTTB), Bamako BP 1805, Mali; 2Sorbonne Université, INSERM, Institut Pierre Louis d’Epidémiologie et de Santé Publique (iPLESP), for Department of Virology, Assistance Publique-Hôpitaux de Paris (AP-HP), Saint-Antoine Hospital, F-75012 Paris, France; 3School of Laboratory Medicine and Medical Sciences, University of KwaZulu-Natal, Durban 4041, South Africa; 4Institute for Global Health, Northwestern University, Chicago, IL 60208, USA; 5Hôpital Régional de Tombouctou, Timbuktu, Mali; 6Department of Pediatrics, Faculty of Medicine, Ain Shams University, Cairo 11566, Egypt; 7Department of Neurology, University of Chicago, Chicago, IL 60637, USA

**Keywords:** prevalence, hepatitis B virus, West Africa, risk factors, HIV, children

## Abstract

While Hepatitis B virus (HBV) and the human immunodeficiency virus (HIV) are endemic in West Africa, the prevalence of HBV/HIV coinfection and their associated risk factors in children remains unclear. In this review, we sought to assess HBsAg seroprevalence among 0- to 16-year-olds with and without HIV in West African countries and the risk factors associated with HBV infection in this population. Research articles between 2000 and 2021 that reported the prevalence of HBV and associated risk factors in children in West Africa were retrieved from the literature using the Africa Journals Online (AJOL), PubMed, Google Scholar, and Web of Science databases as search tools. StatsDirect, a statistical software, was used to perform a meta-analysis of the retained studies. HBV prevalence and heterogeneity were then assessed with a 95% confidence interval (CI). Publication bias was evaluated using funnel plot asymmetry and Egger’s test. Twenty-seven articles conducted across seven West African countries were included in this review. HBV prevalence among persons aged 0 to 16 years was 5%, based on the random analysis, given the great heterogeneity of the studies. By country, the highest prevalence was observed in Benin (10%), followed by Nigeria (7%), and Ivory Coast (5%), with Togo (1%) having the lowest. HBV prevalence in an HIV-infected population of children was (9%). Vaccinated children had lower HBV prevalence (2%) than unvaccinated children (6%). HBV prevalence with a defined risk factor such as HIV co-infection, maternal HBsAg positivity, undergoing surgery, scarification, or being unvaccinated ranged from 3–9%. The study highlights the need to reinforce vaccination of newborns, screening for HBV, and HBV prophylaxis among pregnant women in Africa, particularly in West Africa, to achieve the WHO goal of HBV elimination, particularly in children.

## 1. Introduction

Hepatitis B virus (HBV) infection remains a major cause of acute and chronic liver disease with significant associated morbidity and mortality, worldwide. The World Health Organization (WHO) estimated that 296 million people were living with chronic HBV infection in 2019, with 1.5 million new infections each year and 887,000 deaths due to chronic HBV infection (CHB) [1]. The risk of chronic infection after exposure to HBV depends on age at time of infection, with a 90% risk when infection occurs in infancy and <10% risk when infection occurs in immunocompetent adolescents and adults [2]. Sub-Saharan Africa (SSA) is one of the highest endemic regions with an estimated HBV surface antigen (HBsAg) sero-prevalence (heretofore referred to as HBV prevalence) of more than 8% [3]. In SSA countries, including those in West Africa, the main sources of transmission are mother-to-child during delivery and horizontal during early childhood through close interaction with infected household contacts [4]. Globally, there are 2 million estimated new HBV infections in children < 5 years of age [1], resulting in HBV prevalence of 5–8% in children in SSA, with, for example, a reported 5.8% of five-year-old West African children being infected [5].

HBV burden is particularly high in HIV-endemic areas. HBV prevalence in the people living with Human Immunodeficiency Virus/Acquired ImmunoDeficiency Syndrome, HIV/AIDS is estimated to be more than 7% [6,7,8]. Given similar transmission mechanisms of both HBV and HIV, HBV/HIV co-infection is relatively common in HIV-endemic areas in SSA, in both adults and children. The Joint United Nations Program on HIV/AIDS (UNAIDS) estimated, in 2019, that 1.7 million children were living with HIV worldwide [9]. Few studies, however, have focused on HIV/HBV co-infection in the pediatric population. Although, HBV infection is usually asymptomatic during childhood, co-infection poses a particular challenge as the disease progresses rapidly towards chronicity with increased risk of mortality from cirrhosis or hepatocellular carcinoma (HCC) [10,11,12]. In addition, children are highly vulnerable to HIV infection and are at higher risk of anti-retroviral therapy (ART) failure compared to adults, especially in resource-limited settings [13,14].

Although HIV treatment significantly reduces vertical transmission, some residual risk exists when the status of HIV and HBV are not known during pregnancy. In West Africa, the average childbearing age ranges from 18.8 to 21.8 years and HBV viral load levels are often high in this age group due to immune tolerance [8,15]. In addition, hepatitis B e-antigen (HBeAg) status and high HBV viral load during pregnancy are established risk factors for perinatal transmission [14,15,16]. However, HBV is a vaccine-preventable disease with 95% of properly vaccinated children being well protected. The hepatitis B vaccine (HepB) is recommended for all infants at birth, and for children and adults at high risk [17].

Population-based HBsAg sero-surveys have been recommended as a monitoring tool for the impact of HepB vaccination programs in areas of high and moderate endemicity. However, data on chronic HBV prevalence, HIV co-infection, and other associated risk factors in children in West Africa are limited, particularly following the introduction of HBV vaccination programs. The absolute number of children chronically infected with HBV is not known. Yet, for evaluating national vaccination programs and national disease prevention and control efforts in West Africa and SSA, it is critical to understand the current prevalence of HBV infection.

We, therefore, conducted a systematic review of published studies, between 2000 and 2021, to assess the prevalence in West Africa of HBsAg sero-positivity among persons aged 0 to 16 years, with and without HIV infection, to determine the prevalence of HBV and HIV/HBV co-infection and other risk factors associated with HBV infection in this population.

## 2. Materials and Methods

### 2.1. Inclusion Criteria and Exclusion Reasons

Original research articles in peer-reviewed journals with full-text in English and available online, published between 2000 and 2021, with a clear and concise description of the sample types, size, and methods used to assess HBV sero-prevalence were included. We focused on full-text articles that reported the study design of HBsAg testing to assess HBV sero-prevalence in persons with or without HIV co-infection, conducted in West African countries, and included children, aged 0 to 16 years, in the sample.

Studies conducted among West African populations residing outside Africa were excluded. Systematic reviews, commentaries and editorials, case report studies, surveillance reports, conference abstracts, animal studies, and articles describing the sero-prevalence of HBV only among adult populations were excluded. Studies with insufficient or inaccessible data, pre-prints, studies with a sample size < 100, and studies that did not describe their sampling technique or that used purposive sampling were excluded.

### 2.2. Data Sources and Study Screening Strategies

A systematic search, using key search words, was conducted on Africa Journals Online (AJOL), PubMed, Google Scholar, and Web of Science databases, supplemented by a manual search of retrieved references. The keywords used were HBV prevalence, children, HBV, HBV risk factors, HIV co-infection, and the names of the 14 countries of West Africa (Benin, Burkina Faso, Côte d’Ivoire or Ivory Coast, Cape Verde or Capo Verde, Gambia, Ghana, Guinea, Guinea-Bissau, Liberia, Mali, Mauritania, Niger, Nigeria, Senegal, Sierra Leone, and Togo). Prior to screening, all identified articles were imported into Covidence, a systematic review management tool [18]. Two reviewers (D.B.F. and A.M.S.) independently screened the titles and abstracts for study eligibility and full-text review. Reviewers then independently read the full text and selected studies for inclusion. After each step of the review, the two reviewers met and reached consensus on the final selection of studies for inclusion.

### 2.3. Description of the Study Area

West Africa is comprised of 14 countries (see above) and has one of the world’s fastest-growing populations, with more than 420 million people, equivalent to 5% of the total world population and with a median age of 18 years [19].

### 2.4. Data Extraction

After full-text review, the two reviewers, individually, extracted data from all eligible and retained articles, using the Covidence tool. Any differences observed by a reviewer in data extractio, were reconciled prior to conducting the meta-analysis. In the event of disagreement, a third reviewer (MIK) was consulted to establish consensus. Information extracted from the articles included: sociodemographic characteristics, sample size, the prevalence of HBsAg, risk factors including HIV co-infection, mother’s HBsAg status, HBV contact(s) in the family, blood transfusion, circumcision, or other scarification, surgery, and receipt of HepB vaccine, including a birth dose.

### 2.5. Quality Assessment

The Newcastle–Ottawa scale (NOS) for cross-sectional studies quality assessment tool was used to assess the quality of each study [20]. The tool includes multiple sections. The first section focuses on the sample selection of each study. The second section deals with the comparability of the study including type of study, diagnostic method, and year of publication. The last section focuses on the statistical analysis of each study. A total NOS score ranges from 0 to 10. Study scores of 9–10 points are considered very good, 7–8 as good and 5–6 as satisfactory.

### 2.6. Data Analysis

StatsDirect, a statistical software (Version 3.0.0, StatsDirect Ltd., Cheshire, UK) was used to conduct the meta-analysis of the proportions of HBsAg in the retained studies [21]. Individual study proportions were assessed with a 95% confidence interval (CI) and the pooled effect. Sources of variation among studies were assessed by sub-group analysis, using the following grouping variables: study setting (hospital/other health care setting versus community setting), year of study, year of publication, and the prevalence of HBsAg among children, aged 0–16 years, stratified by “with” group risk factors, “without” group risk factors, and individual risk factors including HIV status, with HBsAg positive mother, and HBV vaccination including birth dose status.

We also assessed the study type, country, sample type, and testing method. Heterogeneity across the studies was assessed by the Quoran (Q) statistic test and the I^2^ statistic. High heterogeneity (I^2^  >  73% and *p* het  <  0.05) represents the percentage of total variation across studies, attributable to heterogeneity rather than to chance [22]. Sources of heterogeneity were analyzed through sub-group analysis and one sensitivity analysis of HIV/HBV co-infection.

Publication bias was evaluated using funnel plot asymmetry and Egger’s test [23]. In this test, *p*  >  0.05 indicates an absence of evidence of publication bias (not significant). A random-effects model (REM) was used to pool HBV prevalence. Some representative results of the REM are presented graphically using forest plots.

We also assessed publication bias analysis in different subgroups. A sensitivity analysis was carried out by excluding 2 studies likely to strongly bias the results of HBV prevalence. Analyses of HBV prevalence were assessed with a 95% CI and *p* < 0.05 was considered as significant. All calculations were done using the StatsDirect software version 3.

## 3. Results

### 3.1. Study Selection

The databases and other literature search tools yielded a total of 1162 articles. Figure 1 summarizes the flow chart for the selection of studies. Duplicate articles (n = 713), due to the diverse search tools used, were removed. Titles and/or abstracts were first used to screen the articles, according to the systematic review and meta-analysis criteria. After screening, 157 articles were further excluded because of a sample size < 100 and/or not within the publication period. However, data were extracted from four general population studies with more than 100 children and adolescents in each of the studies [24,25,26,27]. Of the 278 articles fully screened, 253 were excluded, based on the inclusion and exclusion criteria, as described in the Methods section. A total of 27 articles met the eligibility criteria for the meta-analysis. The included studies were conducted in seven of the fourteen West African countries, Nigeria (thirteen articles), Senegal (five articles), Burkina Faso and Ghana (three articles each), Benin, Togo, and Ivory Coast (one article each), as shown in Table 1.

### 3.2. Characteristics of the Systematic Review Studies

Four were cohort studies, nineteen were cross-sectional, three were retrospective cohort studies, and one was case-control. A total of 11,304 children, aged 0 to 16 years, were included in the retained studies, with studies from Nigeria accounting for most children (3747), of which, 270 were HbsAg positive (270/3747 = 7.2%), followed by Senegal with 3417 children and 80 HBsAg positive, (80/3417 = 2.3%), Burkina Faso, 2383 children (103/2383 = 4.3%), Ghana, 1661 (19/1161 = 1.6%), Ivory coast, 285, (15/2825.3%), Togo, 210 (3/210 = 1.4%) and Benin 103 (10/103 = 9.7%), (Appendix A). We observed a decrease in the prevalence of HBV between 2000 and 2021. HBV prevalence by country and study periods calculated by the StatsDirect software are presented in Figure 2 and Figure 3, respectively.

Multiple techniques were reported to detect HBsAg, with direct enzyme-linked immunoassay test (ELISA) being the most common, followed by rapid diagnostic assays (RDT) using serum or plasma serological markers of HBV.

Nineteen (n = 19), 70% of the studies were conducted in hospital settings and eight (30%) in community settings. Most of the studies involved children only, although we extracted data from four studies of the general population that included >100 children, 0–16 years old [24,25,26,27].

Nine (=9), 33% of studies included persons living with HIV (PLWH), including a total of 2317 children, and three studies included high-risk children born to HBsAg-positive mothers. Most studies were conducted in urban areas, with only four studies being in mixed urban and rural areas. Seven (n = 7), 26% of studies included HepB vaccine status; however, most studies were missing vaccination information (Appendix A).

### 3.3. Overall Pooled HBV Prevalence in Children in West Africa and Publication Bias

HBV prevalence among children, 0–16 years old, varied widely in West Africa (Figure 4A). Crude overall HBV prevalence in the pooled sample of 11,304 children was 5% (95%, CI 4–7%). The seroprevalence data are presented in the random effect model because of the substantial heterogeneity of HBV prevalence across the studies (I^2^ > 94.2%, 95% CI = 93% to 95.1%). Egger’s test was significant (*p* < 0.001) for HBV prevalence, suggesting publication bias in the studies (Figure 4B).

### 3.4. Sensitivity Analysis

To further assess the strength of the HBV prevalence results, we conducted a leave-two-out sensitivity analysis by removing two studies that included two high-risk populations: a study of HBV-HIV co-infected children with high HBV prevalence [37] and a study of infants born to HBsAg positive mothers [27]. The overall HBsAg positive pooled prevalence rate was 5% with heterogeneity prior to the exclusion of these two studies (I^2^ of 94.2%, 95% CI = 93% to 95.1%, *p* < 0.0001). However, after the exclusion, the pooled prevalence rate only dropped to 4% (95% CI 3% to 6%) with heterogeneity (I^2^ of 91.2%, 95% CI = 88.8% to 92.9%), indicating that the pooled HBV prevalence was not affected by the two studies and that the results are robust (Appendix A).

### 3.5. HBV Prevalence in Children with or without Risk Factors

Overall, HBV prevalence in children ranged from 4% (95%CI: 0.03 to 0.06) in children without risk factors, (Figure 5A) to 9% (95%CI: 5 to 15%) in children with at least one risk factor, such as HIV/HBV co-infection or being born to an HbsAg positive mother (Figure 5B).

A funnel plot of HBV prevalence in children with and without HIV shows a not strictly symmetrical display of the prevalence reported by the individual studies (Figure 3). However, the random effects model (DerSimonian Laird) suggests that there is evidence of publication bias, as revealed by the Egger’s test, with a bias = 4.447819 (95% CI = 3.043189 to 5.852449) and *p* < 0.0001.

### 3.6. Risk Factors of HBV Infection in Children

An analysis of children by individual risk factor shows HBV prevalences of 12%, 8%, 5%, and 1% in children born to HBsAg positive mothers, HBV/HIV co-infected, unvaccinated with a birth dose, and vaccinated, respectively, as shown in Figure 6.

### 3.7. HBV Prevalence in Hospital and Community Settings

Subgroup analysis showed higher HBV prevalence in studies conducted in hospital settings [7% (95% CI. 4 to 11%)] compared to studies conducted in community settings among children without any risk factors [3% (95%CI: 2 to 4%)] (Figure 7A,C).

Community setting studies had an Egger Bias of 2.685631 (95% CI = 1.00063 to 4.370632) (*p* = 0.0094), corresponding to low heterogeneity between studies, compared to an Egger Bias of 5.116854 (95% CI = 2.916234 to 7.317473) (*p* = 0.0002) for hospital setting studies, corresponding to a high heterogeneity (Figure 7B,D).

### 3.8. HBV Prevalence by HIV Status in Children

HBV/HIV co-infected children had an overall HBV prevalence of 9% (95% CI: 4 to 15%), with an Egger bias of 6.253247 (95% CI = 2.533005–9.97349; *p* = 0.0054), corresponding to low heterogeneity between studies, as opposed to studies of HBV mono-infected children who had an overall HBV prevalence of 4% (95% CI, 2 to 6%) and an Egger bias of 3.7865 (Figure 8).

#### HBV Prevalence among Vaccinated and Unvaccinated Children

HBV prevalence was still high in studies of unvaccinated children [6% (95%, CI: 4–8%)] (Figure 9A) compared to studies of vaccinated children [2% (95%, CI: 1–3%)] (Figure 9B). Due to a lack vaccination status data in most studies, HBV prevalence caused by natural infection (anti-HBs + anti-HBc) could not be evaluated.

## 4. Discussion

A systematic review of HBV prevalence in children in West Africa aged 0–16 years yielded 27 studies meeting inclusion criteria for a meta-analysis. The studies, however, only included data from seven of West Africa’s fourteen countries. Most studies were cross-sectional; three-quarters used the Enzyme-Linked Immunosorbent Assay (ELISA) as the HBV diagnostic tool, and most were conducted in a hospital setting (18/27) in an urban area. Indeed, most healthcare facilities in West Africa are in urban areas.

Overall, HBV prevalence was 5%, based on the random analysis, given the heterogeneity of the studies. This meta-analysis revealed that HBV prevalence in children in West Africa is moderate, according to the WHO’s criteria for HBV endemicity [51]. HBV prevalence by country ranged from 1% in Togo to 10% in Benin. The prevalence was also variable inside the same country, such as in Nigeria or Ghana, for which there are a high number of published data available. Differences in health system resources across West African countries, although already generally limited in all of West Africa, may still explain some of the discrepancies. In fact, some countries such as Mali and Benin lack resources to screen pregnant women or provide vaccine doses at birth for newborns. Findings from a study with timely dose of hepatitis B Birth dose (HepB-BD) in Africa demonstrated the variability of birth dose implementation and the challenges countries face in immunizing babies [52]. In the absence of efforts for preventing mother-to-child transmission, it is anticipated that HBV prevalence in children in these countries would stay higher and fall short of the WHO targets.

Due to the significant burden of HBV-related diseases in SSA, where >8% of the general population is chronically infected, HBV infection is still ubiquitous across West Africa. This can impact HBV prevalence in these countries [53] and the prevalence is around 5% among pregnant HBV/HIV co-infected women [54]. There has been a significant decrease in the prevalence of HBsAg among children in West Africa since the year 2000. This is clearly demonstrated by comparing the 12% HBsAg prevalence in children and adolescents < 19 years old in West African SSA in 1990 [51], and the prevalence of HBV infection in children at 2.53% in 2019, amounting to 360,000 infected children each year [55]. We believe that the decrease may be related to the introduction of infant HBV vaccination programs in most African countries and including the HepB-BD in a few countries.

Studies conducted in hospital settings showed higher prevalence of HBV infection (7%) and this is likely related to the studies being conducted on sick children, who are also tested at higher rates than non-hospitalized children. By contrast, healthy children in the community setting had an HBV prevalence of only 3% and may represent early success of the newborn HepB vaccination programs. Prevalence varied from 3% to 9% among infants with HBV/HIV co-infection, those born to mothers who tested positive for HBsAg, and those who had other risk factors, such as having undergone surgery, scarification, or not receiving vaccinations. Other risk factors were not available in many studies. In one Nigerian study, risk factors such as previous history of jaundice (*p* = 0.26), blood transfusion (*p* = 0.24), past history of surgery (*p* = 0.47), or scarification marks (*p* = 0.17) were not associated with HBV prevalence [26].

HIV and HBV infections have many similarities, such as common routes of transmission, high prevalence in certain geographical regions, same at-risk groups, and the risk of mother-to-child transmission. All these factors contribute to a significant association between HBV and HIV co-infection in the pediatric population [56,57]. We found an overall HIV/HBV co-infected prevalence of 9%. However, in a sensitivity analysis, we removed one study from Burkina Faso (2001) that reported a co-infection rate of 40%, and yielded a prevalence of 6%, ranging from 1.15% in Nigeria (2021) to 10% in Benin (2015) (Figure 3). The finding from this study is consistent with prior studies. In studies from South Africa, HIV/HBV co-infection prevalence in children ranged from 5–17%, with the higher prevalence occurring in the industrialized towns where mining activity is associated with increased sexual activity. In Nigerian studies, HBV/HIV co-infection prevalence varied by geopolitical region, ranging from 5.8 to 19% [13,29,38,40,58]. Finally, a Tanzanian study of HBV/HIV co-infected children reported 1.2% prevalence [59]. Reported HBV/HIV co-infection prevalence in children has been somewhat higher in Nigeria (7.8%) [29], 10.4% in Zambia [60], and 12.1% in Ivory Coast [61]. The generational effect benefits of HepB vaccination may be a factor in the relatively lower HBV-HIV co-infection in later studies compared to earlier studies. It is also important to note that HIV/HBV co-infection considerably increases the risk of mother-to-child transmission, if the mother is infected and untreated.

Perinatal and childhood acquisition of HBV not only leads to increased risk of chronicity but also strongly predict worse long-term outcomes for liver cirrhosis and hepatocellular carcinoma [16]. HIV/HBV co-infection in childhood further places children at high risk for associated morbidity and mortality, similar to adults, hence the urgent need to implement HepB vaccination at birth, routine screening, and follow-up. In fact, despite more than two decades since the introduction of HepB vaccination programs, the overall prevalence of HBV infection still remains high in many settings in SSA [53,62,63].

The difference in HBV prevalence infection between vaccinated and unvaccinated children is still substantial (2% versus 6%). While there is a residual possibility of HBV infection despite newborn HepB vaccination, often, it is unclear if infection occurred before or after completion of the HepB vaccine series [41]. In SSA, horizontal transmission in children, aged 6 months to 5 years, is common due to close interactions with infected household contacts and playmates. We examined the impact of HepB newborn vaccination compared to the evolution of the epidemic in each country. In Nigeria, overall HBV prevalence was 7% and a case-control study found that HBsAg prevalence was significantly lower among vaccinated (1.4%) compared with unvaccinated (4.8%) children [49]. In Senegal and Ghana, where all the studies were conducted after newborn HepB vaccination had been introduced, overall HBV prevalence was only 2%. In Togo, one in ten women of childbearing age was infected with HBV, but less than 2% of infants under five years of age who received the HepB vaccine at birth were infected with HBV. These results are similar to other studies that found significant reductions in HBsAg positivity post-vaccination with a protective efficacy of between 67–94% [64,65]. Such a high level of vaccine efficacy is likely to positively impact and prevent community transmission of HBV.

In 1992, the World Health Assembly adopted a resolution recommending the introduction of HepB into national immunization programs [66]. In 2016, the WHO and the World Health Assembly established the target of controlling and eliminating HBV worldwide by 2030. The WHO recommends that all infants receive their first dose of HepB as soon as possible after birth, ideally within 24 h, followed by two or three doses of HepB at least 4 weeks apart to complete the vaccination series [67]. Despite this recommendation, HepB vaccination at birth has not been widely implemented in most national vaccination programs in SSA, particularly in West Africa, despite the Expanded Program of Immunization (EPI) [68]. Indeed, in most West African countries, parents must still pay for the first HepB dose, which, at ~$US 8 per dose, is prohibitive, given that 85% of the population in SSA live on less than $5.50/day [69]. In 2021, coverage with timely HepB-BD was 42% globally and only 17% for infants in the WHO African region [70]. A total of 114 countries worldwide had introduced HepB-BD in their routine immunization schedule [71]. Yet, this number includes only 14 (30%) of 47 countries in the WHO African region [72]. Limited countries have data on Studies of the Effectiveness of HepB-BD Vaccination in Africa with only two studies published to date. In 2001–2002, Ekra and colleagues conducted a nonrandomized controlled trial in four health centers in Abidjan, Cote d’Ivoire [73] and a second effectiveness study was conducted from 2009–2016 in a single center in Tokombéré district, Cameroon [74]. Based on the results of those published studies, the authors assumed a high-level transmission rate in the absence of vaccination and highlighted the benefit of the addition of HepB-BD to the three-dose HepB vaccination schedule for infants. In addition, a residual risk of mother-to-child transmission of hepatitis B virus infection despite timely birth-dose vaccination in Cameroon has been reported. In addition, studies from Hawaii, Taiwan, and China provide assurance that routine infant HepB immunization beginning with HepB-BD vaccination is a highly effective public health strategy that will progressively protect generations to come from HBV-related liver disease, HCC, and premature mortality [75,76].

The universal vaccination strategy implemented in Thailand provides evidence of the effect of newborn HepB vaccination (HepB-BD) in eliminating HBV infection [77]. As more African countries seek to implement HepB-BD, attention to disparities in implementation need to be addressed particularly in rural and underprivileged settings [63]. Maternal education and community engagement are essential to scale up HepB-BD in SSA through the Global Alliance for Vaccines and Immunisation (GAVI). A study in Nigeria demonstrated that missed doses were largely avoided when staff completed a vaccine checklist before releasing mother–child pairs [78]. There must also be a focus on offsetting the costs of distributing the HepB-BD throughout SSA.

In endemic regions of Africa and Asia, in-utero infection of the fetus, vertical transmission, constitutes the main mode of HBV transmission [79]. A meta-analysis indicated that maternal viral load was an important risk factor for mother-to-child transmission (MTCT) in HBeAg-positive mothers, and maternal viral load was dose-dependent with HBV MTCT incidence [80]. Other studies have shown that a high viral load in HBsAg-positive mothers can lead to vaccination failure in the newborn, even if combined immunoglobin treatment and vaccination at birth are delivered [81].

HBsAg screening for all pregnant women is critical. Focusing on HBsAg positive women and providing prophylactic treatment to women with high viral loads can be an effective approach to reduce transmission to the infant [82]. A Senegalese study of HBV in children born to HIV-positive mothers showed a low rate of HBsAg (2.6%) if the mother was treated with lamivudine (3TC) or tenofovir (TDF), compared to 7.9% if the mother was untreated [46]. Another Senegalese study showed low HBV prevalence, but only 56% of children had a sero-protective level >10 UI/L [34]. It is, therefore, strongly recommended to vaccinate children and to adhere to the necessary doses to protect them against HBV infection.

### Limitations

This meta-analysis has several limitations, particularly the substantial heterogeneity of eligible studies. Prior HBV prevalence meta-analyses in Africa have also had high levels of heterogeneity [83,84]. Data about children, aged 0–16 years, from four general population studies were used and were the only data available from several countries. However, it was challenging to analyze risk factors for children separately from adults in these studies. The quality of the studies varied between countries, as well as within the same country. Studies showed that protective levels of HbsAb antibodies decrease with age after vaccine introduction [30,43]; however, most of the studies included in this review did not report prevalence by age group.

## 5. Conclusions

In conclusion, this study using the most recent data available estimated HBV prevalence in children aged 0–16 years in West Africa, revealing a decrease in prevalence over the past two decades; yet, a persistently high prevalence in high-risk child populations still exists. The studies are robust and cover periods before and after the introduction of HepB vaccination in different West African countries. The meta-analysis identified wide variation in HBV prevalence, depending on the country, as well as in subgroups of HBV/HIV co-infected children, children born to HBsAg-positive mothers, and vaccinated or unvaccinated children. Overall, pooled prevalence and pooled subgroup prevalence remain moderate in West Africa and reaffirm the likely impact of newborn HepB vaccination. The study reinforces the need to implement HepB-BD and screening and prophylaxis of HBV in pregnant women. These interventions are essential to prevent mother-to-child transmission in order to achieve the WHO goal of targeted HBV elimination.

## Figures and Tables

**Figure 1 ijerph-20-04142-f001:**
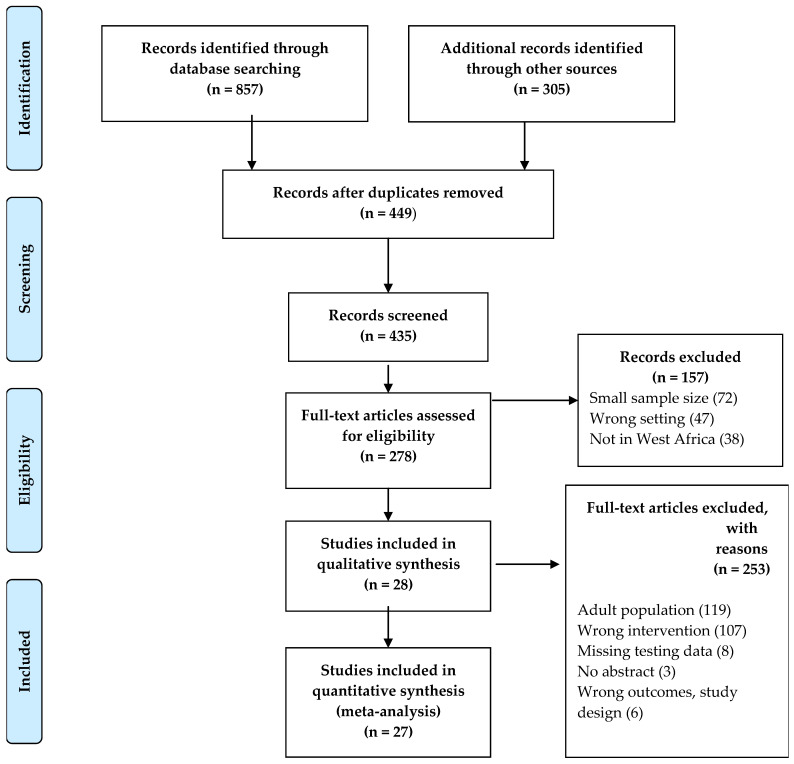
Study Eligibility Flow Diagram.

**Figure 2 ijerph-20-04142-f002:**
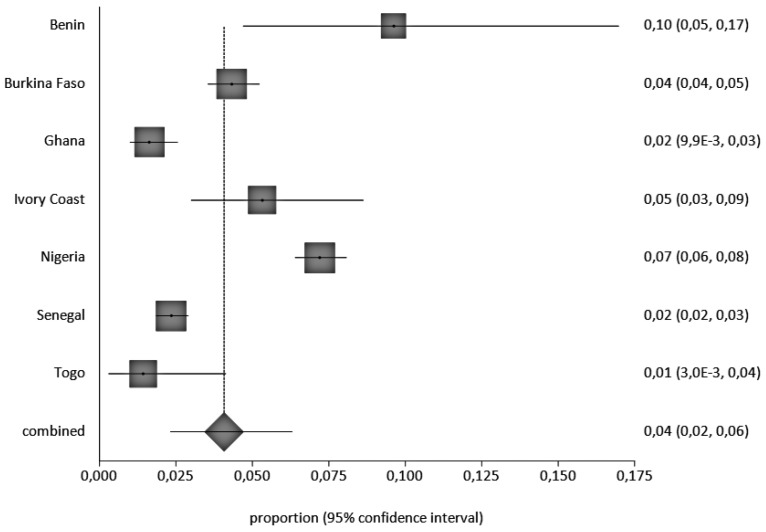
Overall HBV Prevalence in Children, 0–16 Years Old, in West Africa, Pooled by Country. A black square represents the HBV prevalence in the forest plot. The square position represents the prevalence.

**Figure 3 ijerph-20-04142-f003:**
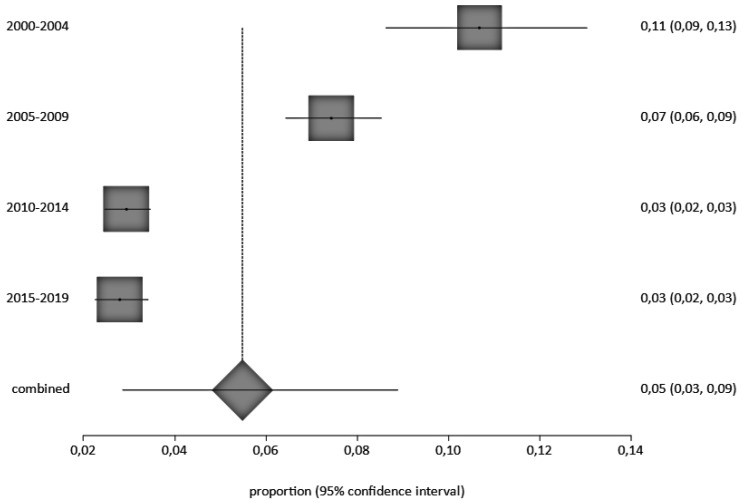
Forest Plot of Pooled HBV Prevalence by Year of Study Interval. A black square represents the HBV prevalence in the forest plot. The square position represents the prevalence.

**Figure 4 ijerph-20-04142-f004:**
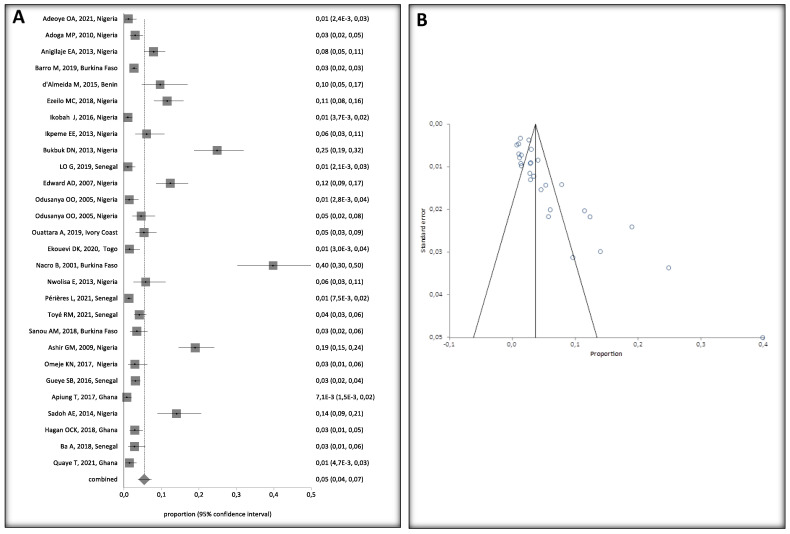
(**A**) Forest Plot of Global HBV Prevalence in Children in West Africa, 2000 and 2021. A black square represents the HBV prevalence in the forest plot. The square position represents the prevalence in each study in the meta-analysis [24,25,26,27,28,29,30,31,32,33,34,36,37,38,39,40,41,42,43,44,45,46,47,48,49,50]. (**B**) Bias Assessment Plot.

**Figure 5 ijerph-20-04142-f005:**
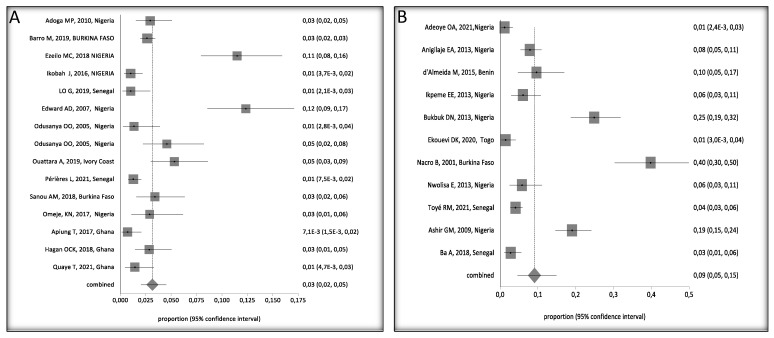
Forest Plots of HBV Prevalence in Children 0–16 Years Old in West Africa, (**A**) Without Risk Factors and (**B**) With Risk Factors [24,25,26,27,28,29,30,31,32,33,34,36,37,38,39,40,41,43,44,45,47,48,49,50]. A black square represents the HBV prevalence in the forest plot. The square position represents the prevalence in each study in the meta-analysis.

**Figure 6 ijerph-20-04142-f006:**
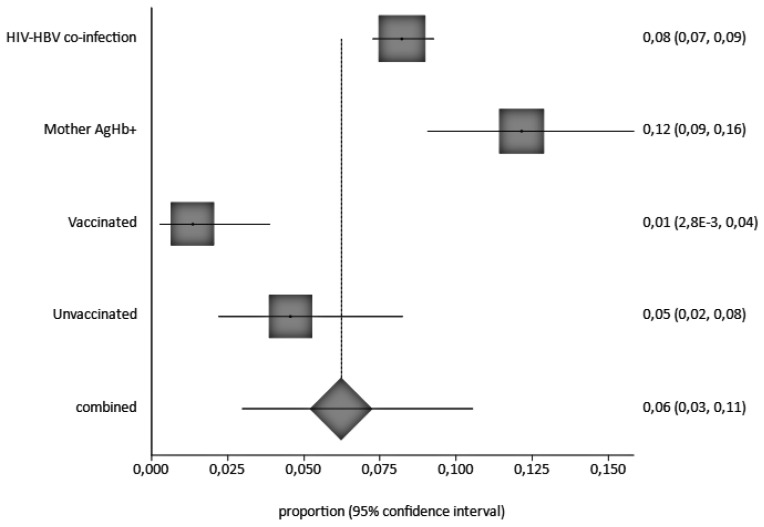
Pooled HBV Infection Prevalence by Risk Factor A black square represents the HBV prevalence in the forest plot. The square position represents the prevalence.

**Figure 7 ijerph-20-04142-f007:**
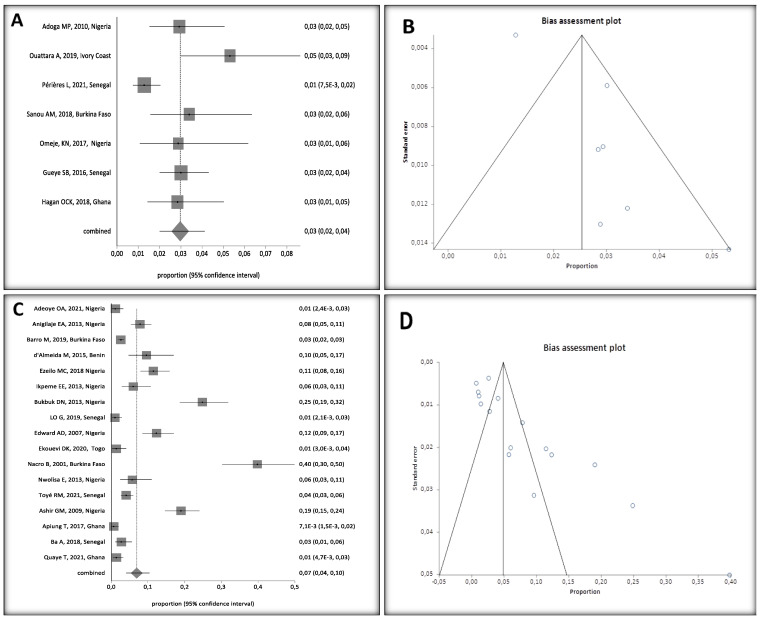
(**A**) Forest Plots of HBV Prevalence in Children, 0–16 Years Old, in West Africa, Conducted in Community Settings, (**B**) Heterogeneity in Community Setting Studies, (**C**) Forest Plots of HBV Prevalence in Hospital Settings, and (**D**) Heterogeneity in Hospital Setting Studies [24,25,26,28,29,30,31,32,33,34,36,37,38,39,40,41,43,44,45,46,47,48]. A black square represents the HBV prevalence in the forest plot. The square position represents the prevalence in each study in the meta-analysis.

**Figure 8 ijerph-20-04142-f008:**
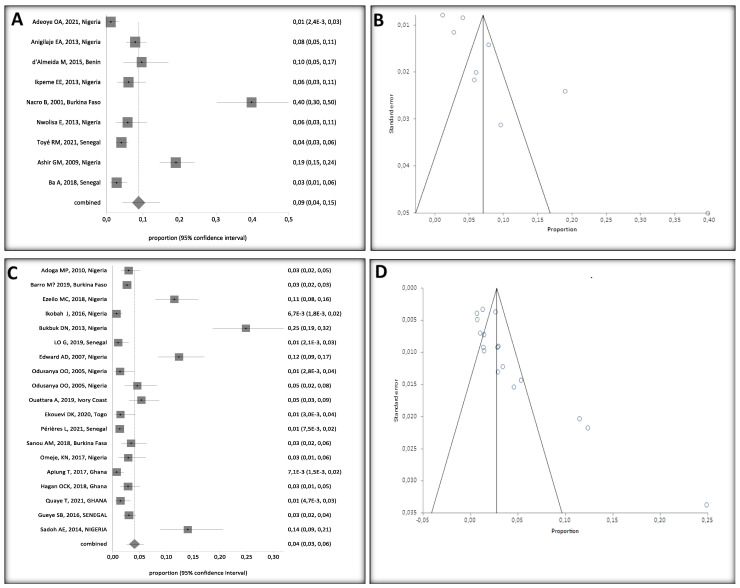
(**A**) Forest Plots of HBV/HIV Co-Infected Children, (**B**) Bias Assessment of Co-Infected Children, (**C**) Forest Plots of HBV Mono-Infected Children, (**D**) Bias Assessment of HBV Mono-Infected Children [24,25,26,27,28,29,30,31,32,33,34,36,37,38,39,40,42,43,44,45,47,48,49,50]. A black square represents the HBV prevalence in the forest plot. The square position represents the prevalence in each study in the meta-analysis.

**Figure 9 ijerph-20-04142-f009:**
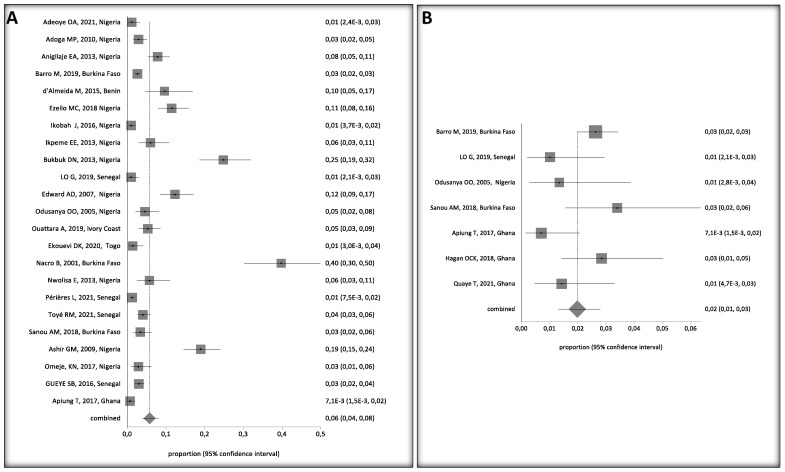
(**A**) HBV Prevalence Among Unvaccinated and (**B**) Vaccinated Children [24,25,26,27,28,29,30,31,32,33,34,36,37,38,39,40,41,43,45,46,47,49]. A black square represents the HBV prevalence in the forest plot. The square position represents the prevalence in each study in the meta-analysis.

**Table 1 ijerph-20-04142-t001:** Characteristics of Systematic Review Studies and Meta-Analysis of HBV Prevalence in Children in West Africa.

1st Author [Reference]	Study Setting	Study Year	Publication Year	Country	Study Type	Sample Size	Number AgHBs+	Prevalence of AgHB+ (%)	Diagnostic Method	Sample Type	Score Quality
Studies in Hospital Settings											
Adeoye OA [28]	Urban	2012	2021	Nigeria	Cross-sectional	261	3	1.1	ELISA	SERA	7
Anigilaje EA [29]	Urban	2008–2012	2013	Nigeria	Cross-sectional	395	31	7.8	ELISA	SERA	10
Barro M [30]	Urban	2013	2019	Burkina Faso	Cohort	2015	53	2.6	ELISA	PLASMA	9
d’Almeida M [31]	Urban	2014	2015	Benin	Cross-sectional	104	10	9.6	RDT	PLASMA	9
Ezeilo MC [32]	Urban	2017	2018	Nigeria	Cross-sectional	270	31	22.5	ELISA	PLASMA	7
Ikpeme EE [33]	Urban	2010–2011	2013	Nigeria	Cohort	166	10	6.0	ELISA	SERA	9
Bukbuk DN [27]	Urban	2009–2010	2016	Nigeria	Cohort	177	44	24.9	ELISA	SERA	6
LO G [34]	Urban	2016	2019	Senegal	Cross-sectional	295	3	1.1	ELISA	SERA	9
Edward AD [35]	Urban	2004	2007	Nigeria	Retrospective	251	31	12	ELISA	SERA	7
Ekouevi DK [36]	Urban	2017	2020	Togo	Cross-sectional	210	3	1.3	RDT	SERA	7
Nacro B [37]	Urban	2001	2001	Burkina Faso	Cross-sectional	103	41	39.8	RDT	SERA	8
Nwolisa E [38]	Urban	2010	2013	Nigeria	Cross-sectional	139	8	5.8	RDT	SERA	6
Toyé RM [39]	Urban	2015	2021	Senegal	Retrospective	613	25	4.1	RDT	SERA	8
Ashir GM [40]	Urban	2007	2009	Nigeria	Cross-sectional	284	54	2.8	ELISA	SERA	7
Apiung T [41]	Urban	2012–2013	2017	Ghana	Cross-sectional	424	3	0.05	ELISA	PLASMA	7
Sadoh AE [42]	Urban	2011	2014	Nigeria	Cross-sectional	150	21	13.9	ELISA	SERA	8
Hagan OCK [43]	Urban	2012–2013	2018	Ghana	Cross-sectional	387	11	2.8	ELISA	SERA	7
Ba A [44]	Urban	2013–2015	2018	Senegal	Cross-sectional	252	7	2.8	ELISA	SERA	7
Quaye T [45]	Urban	2019	2021	Ghana	Cross-sectional	350	5	1.4	ELISA	SERA	8
Studies in Community Settings											
Gueye SB [46]	Urban	2007–2012	2016	Senegal	Retrospective	930	28	3.0	ELISA	TOTAL BLOOD	7
Omeje KN [26]	Urban	2010–2011	2017	Nigeria	Cross-sectional	208	6	2.8	RDT	SERA	5
Sanou AM [47]	Rural	2015	2018	Burkina Faso	Cross-sectional	265	9	3.4	RDT	SERA	6
Périères L [48]	Rural	2018–2019	2021	Senegal	Cross-sectional	1327	17	1.2	ELISA	SERA	10
Ouattara A [25]	Urban	2006	2019	Cote d’Ivoire	Cross-sectional	282	15	5.3	ELISA	SERA	9
Odusanya OO [49]	Rural/Urban	2001	2005	Nigeria	Case control	223	3	1.2	ELISA	SERA	8
Odusanya OO [49]	Rural/Urban	2001	2005	Nigeria	Case control	219	10	4.6	ELISA	SERA	8
Ikobah J [50]	Urban	2014	2016	Nigeria	Cross-sectional	595	6	0.6	RDT	SERA	7
Adoga, MP [24]	Urban	2008–2009	2010	Nigeria	Cohort	409	12	2.9	ELISA	SERA/PLASMA	8

## Data Availability

The data presented in this study are available on request from the corresponding author.

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
