# Peer review of "Hepatitis B Virus in West African Children: Systematic Review and Meta-Analysis of HIV and Other Factors Associated with Hepatitis B Infection"

_ijerph, 2023, doi:10.3390/ijerph20054142_

Round 1

Reviewer 1 Report

Authors investigated the prevalence of HBV/HIV coinfection and their associated risk factors in children in West African. This study is well-designed, but some issues need to be addressed.

1.      Many typos are found in the manuscript. Proofreading is strongly suggested.

2.      The format of the tables is suggested to be adjusted. Three-line tables are recommended.

3.      The figure legends of all figures should be more detailed. Also, the quality of figures needs to be improved.

4.      Only data from 7/14 West Africa countries were included. Is there any possibility that it may lead to any bias in this study?

5.      Figure 2 showed that there is a difference in HBV prevalence in children among the 7 countries. What leads to the difference? Authors should discuss about it.

6.      Page 12, Line 337: Authors claimed that there has been a significant decrease in the prevalence of HBsAg among children in West Africa. The prevalence in different countries should be discussed. Also, references of “the introduction of some HBV vaccination in most African countries” should be cited.

7.      Page 13, Line 345: Authors indicated that “healthy children in the community setting had an HBV prevalence of only 3% and may represent early success of newborn HepB vaccination programs”. Is there any difference in newborn HepB vaccination programs between children in hospitals and in communities? Why sick children should be tested at higher rates of HBV infection than non-hospitalized children?

8.      Page 13, Line 348: Authors cited a Nigerian study to indicate that other risk factors were not related with HBV prevalence. Is this study included in this review? Do other studies support this study’s findings?

Author Response

We would like to thank the reviewer for the suggestions/comments which will improve the quality of this review if accepted.

Reviewer 2 Report

Peer-review – International Journal of Environmental Research and Public Health – Feb 2023

Article

Hepatitis B Virus in West African Children: Systematic Review and Meta-Analysis of HIV and Other Factors Associated with Hepatitis B Infection

Authors:

  Djeneba B. Fofana, Anou M. Somboro, Mamoudou Maiga, Mamadou I. Kampo, Brehima Diakité, Yacouba  Cissoko, Sally M. McFall, Claudia A. Hawkins, Almoustapha I. Maiga, Mariam Sylla, Joël Gozlan, Manal H. El Sayed, Laurence Morand-Joubert, Robert L. Murphy, Mahamadou Diakité  and Jane L. Holl

Dear authors, your manuscript was reviewed and below are my comments.

Thanks.

Introduction

Line 65: HIV-infected is stigmatizing, prefer People Living with HIV/AIDS

Overall, this is a long introduction. Paragraphs 4-5 could be moved to discussion or even condensed.

Material and Methods

Very well written, with all required details and information for a systematic review and meta-analysis.

Results

Figure 1. It would be good to see the number/percentage among the records excluded after the first screening (e.g., N/% of small sample size, wrong setting…) and the same for full-text articles (add the percentage and correctly align the lines)

Why there is a difference of 01 article between “studies included in qualitative synthesis” and “studies included in quantitative synthesis”?

Table 1. Very informative. Just a detail: for a table, the borders inside the table/for the rows and columns are not recommended. Just the outside borders.

I just missed the analysis of HBV prevalence among children from mothers with HbsAg positivy vs HbsAg negative, as the HbsAg positivity was the strongest risk factor for HBV infection in children.

Discussion

Overall, good. Addressed and discussed the many findings in results.

Conclusion

Great! Summarize the study and address solutions.

Reviewer’s final comment: the authors well developed a systematic review and meta-analysis of HBV prevalence in West Africa. The manuscript is well written. The methods section describes all the (required) steps/procedures for this type of study. I just had some minor comments and suggested editions – addressed above and highlighted in the original manuscript file (attached). I look forward to review this manuscript again.

Thanks,

Author Response

We would like to thank the reviewer for the suggestions/comments which will improve the quality of this article if accepted.

Reviewer 3 Report

Use complete words instead of abbreviations when used for the first time in the manuscript.

In the flow chart include the box of keywords searched above the identification box, page # 5.

Table # 1, must also include the column of % of AgHBs+ .

The author is suggested that the data may be analyzed in from three different angles separately, that is Studies in Hospital Settings, Studies in Community Settings, and Nigeria as a separate unit because the variation in the percentage of AgHBs+ differs a lot. (The articles referred to in table 1, show some major limitations. Nigeria (Adeoye OA[32] 261 /3, Anigilaje EA[33] 395 / 31, Ezeilo MC[36] 270/31. Ashir GM [44] 284/54).

The result portion may consist of the overall presentation of data, Studies in Hospital Settings, Studies in Community Settings, and Nigeria as a separate unit.

A separate paragraph regarding the limitations or shortcomings of the study must be included in the manuscript.

Author Response

(The authors gave the same response as above.)

Round 2

Reviewer 1 Report

The authors have revised the manuscript accordingly and the quality improves a lot. However, some minor points need to be addressed before the publication can be considered. For example, abbreviations should be written in full names the first time they appear in the manuscript (Page 2, AIDS).

Reviewer 3 Report

Recommended, please,